# Superior ferroelectricity and nonlinear optical response in a hybrid germanium iodide hexagonal perovskite

Kun Ding[1,2,4], Haoshen Ye[3,4], Changyuan Su [1,4], Yu-An Xiong[1], Guowei Du[1], Yu-Meng You [1], Zhi-Xu Zhang [1,2] ✉, Shuai Dong [3] ✉, Yi Zhang [2] ✉ & Da-Wei Fu [2] ✉

Abundant chemical diversity and structural tunability make organic–inorganic hybrid perovskites (OIHPs) a rich ore for ferroelectrics. However, compared with their inorganic counterparts such as $BaTiO_3$, their ferroelectric key properties, including large spontaneous polarization ($P_s$), low coercive field ($E_c$), and strong second harmonic generation (SHG) response, have long been great challenges, which hinder their commercial applications. Here, a quasi-one-dimensional OIHP $DMAGeI_3$ (DMA = Dimethylamine) is reported, with notable ferroelectric attributes at room temperature: a large $P_s$ of 24.14 μC/cm$^2$ (on a par with $BaTiO_3$), a low $E_c$ below 2.2 kV/cm, and the strongest SHG intensity in OIHP family (about 12 times of $KH_2PO_4$ (KDP)). Revealed by the first-principles calculations, its large $P_s$ originates from the synergistic effects of the stereochemically active $4s^2$ lone pair of $Ge^{2+}$ and the ordering of organic cations, and its low kinetic energy barrier of small DMA cations results in a low $E_c$. Our work brings the comprehensive ferroelectric performances of OIHPs to a comparable level with commercial inorganic ferroelectric perovskites.

Ferroelectrics with switchable spontaneous electrical polarizations in response to electric fields, having demonstrated great scientific and application values in data storage, sensors, optoelectronics, and versatile micro-electro-mechanical systems (MEMSs)[1–7]. As the most important properties of ferroelectricity: spontaneous polarization ($P_s$), coercive electric field ($E_c$), and Curie temperature ($T_C$), vitally determine the comprehensive performances of ferroelectric materials and their commercial value for applications[8–11]. Inorganic oxide ferroelectrics, e.g., perovskite $BaTiO_3$ and $Pb(Zr, Ti)O_3$, have long been the successful mainstream of commercial applications, thanks to their high $T_C$, large $P_s$, small $E_c$, as well as chemical stability[12–15]. However, their rigid structures, non-passive surfaces, and incompatibility with silicon industry, becomes more and more challenging for

next-generation devices, such as nanoscale MEMSs and flexible electronics.

With the requirements from flexible applications, organic-inorganic hybrid perovskite (OIHP) ferroelectrics have emerged as a promising branch, by taking the advantage of low-cost manufacture, environmentally friendly, ease of fabrication, and excellent film formation[16–21]. OIHP ferroelectrics own rich structural diversity[22–28], including three-dimensional (3D) $ABX_3$-type (pseudo-)cubic structures[29,30], two-dimensional (2D) layered structures[31,32], and even quasi-one-dimensional (1D) $ABX_3$-type hexagonal structures[33]. Different with 3D and 2D OIHPs whose structures are limited by many issues such as tolerance factor, 1D OIHPs do not have strict selections on metal halide anions and organic cations regarding their sizes, shapes,

[1]Jiangsu Key Laboratory for Science and Applications of Molecular Ferroelectrics, Southeast University, Nanjing 211189, China. [2]Institute for Science and Applications of Molecular Ferroelectrics, Key Laboratory of the Ministry of Education for Advanced Catalysis Materials, Zhejiang Normal University, Jinhua 321019, China. [3]Key Laboratory of Quantum Materials and Devices of Ministry of Education, School of Physics, Southeast University, Nanjing 211189, China. [4]These authors contributed equally: Kun Ding, Haoshen Ye, Changyuan Su. ✉e-mail: zhixu@seu.edu.cn; sdong@seu.edu.cn; yizhang1980@seu.edu.cn; dawei@seu.edu.cn

or valences. Thus 1D OIHPs are endowed with great structural freedom to engineer high-performance ferroelectrics with intriguing functionalities. Indeed, some 1D OIHP ferroelectrics have been synthesized and exhibit interesting properties such as large piezoelectric response[34], multi-step nonlinear optical switches[35], photoluminescence, magnetism[36], and circularly polarized luminescence[37]. In particular, two 1D OIHP ferroelectrics [Me₃NCH₂Cl]$MCl_3$ ($M$ = Mn and Cd) have been reported to show large piezoelectric response on par with BaTiO₃[38]. Even though, the comprehensive performances of available OIHP ferroelectrics remain incompetent when competing with conventional ferroelectric oxides. The most serious problems are their small $P_s$ (typically <10 μC/cm²) and relatively high $E_c$ (~100 kV/cm). For comparison, the $P_s$ of BaTiO₃ reaches 26 μC/cm² and its typical $E_c$ is ~10 kV/cm[39].

Screening and modifying organic cations (imidazole, pyrrolidine, quinuclidine, etc.) are efficient strategies to improve the material performance[40]. Because the polarization switching of OIHPs is generally related to the ordering reorientation of organic cations, it appears that small organic cations may be beneficial in reducing the coercive field. Besides, the inorganic framework of $BX_6$ octahedra is also with remarkably compositional diversity. But their contributions to polarization have mostly been omitted since those octahedra formed by common transition metals (i.e., Cd, Cr, Mn, Cu) are generally inactive to ferroelectricity.

Inspired by the aforementioned ideas (Supplementary Fig. 1), we used the small DMA⁺ cation and germanium halide to synthesize a series of OIHP DMAGe$X_3$ ($X$ = Cl, Br, I) (Supplementary Note 1). Remarkably, OIHP DMAGeI₃ exhibits excellent ferroelectricity with $T_C$ = 363 K, a high $P_s$ = 24.14 μC/cm², and a low $E_c$ = 0.8-2.2 kV/cm at room temperature. To our best knowledge, such an experimental $P_s$ is higher than those of most reported pure-organic and inorganic-organic hybrid ferroelectrics, and rivals the popular ferroelectric oxide BaTiO₃. And equally importantly, its $E_c$ value is one order of magnitude lower than that of recently reported inorganic perovskite CsGeI₃ (~40 kV/cm)[41], and two orders of magnitude lower than those of PVDF (~500 kV/cm) and its copolymers[39]. Moreover, it owns a large SHG response, whose intensity is more than ten times stronger than that of KH₂PO₄ (KDP, a commonly used standard system). Our finding brings the comprehensive performances of OHIP ferroelectrics to a comparable level with typical inorganic ferroelectric perovskites.

## Results
### Structural analysis of crystal
Yellow and transparent rod-shaped single crystals of DMAGeI₃ in size of 2 × 2 × 16 mm³ (Fig. 1a) were prepared by hydrothermal method (Supplementary Note 1). The phase purity and thermal stabilities of grown crystals were identified by powder X-ray diffraction (XRD) and thermogravimetric analysis (TGA), which shows good environmental and thermal stability (Supplementary Figs. 2 and 3). Phase transition of DMAGeI₃ was characterized using differential scanning calorimetry (DSC) measurements. As shown in Supplementary Fig. 4, the DSC

curves show a pair of thermal anomalies peaked at 359 K and 363 K in cooling/heating runs, respectively. For convenience, we defined the phase above $T_C$ of 363 K as the high-temperature phase (HTP) and the phase below $T_C$ as the low-temperature phase (LTP). Crystal structures in LTP and HTP were determined by variable-temperature single-crystal X-ray diffraction in the range of 223-373 K (Supplementary Note 2 & Table 1).

The structural analysis discloses that DMAGeI₃ crystallizes in a centrosymmetric space group *Pnan* in HTP, while in LTP it transforms into a polar space group *Pna2₁*. Figure 1b, c show that the two phases are similar except for the DMA orientation and GeI₃ distortion. DMAGeI₃ adopts the hexagonal perovskite structure containing 1D inorganic framework, where the GeI₃ units are linked via the face-sharing GeI₆ octahedra. The cavities between GeI₃ columns are filled by organic DMA cations, which are loosely connected with the inorganic framework via N-H···I and C-H···I hydrogen bonding interactions. In HTP, the organic DMA cations locate at the 2-fold crystallographic rotation axes of lattice and exhibit orientational disorder toward two equivalent directions, making their dipole moments cancel each other (Fig. 1b). In LTP, the DMA cations become ordered with an aligned manner along the *c*-axis (Fig. 1c and Supplementary Fig. 5), and also their positions move significantly (0.38 Å) along the *c*-axis direction compared with those in HTP (Supplementary Fig. 6), both of which induce a net dipole moment in this direction (Fig. 1c). Concomitantly, the Ge-I bond lengths and I-Ge-I angles in LTP also changed obviously (Supplementary Fig. 7 and Supplementary Table 2), although the GeI₆ octahedra are already distorted in HTP. The value of the octahedron distortion parameter Δ and angle variance $\sigma_{oct}$ rise from 0.0096/60.6 (HTP) to 0.0121/69.3 (LTP) respectively, which reflect a more distorted state of GeI₆ octahedra in LTP (Supplementary Table 3).

These structural characteristics, including the orientation ordering of DMA cation, the structural distortion of GeI₃ framework, and the relative moving between these two parts, will contribute to a ferroelectric polarization along the *c*-axis, which will be further analyzed later.

### Characterization of ferroelectricity
According to the above structural analysis, the symmetry breaking occurs with the change of space group from *Pnan* to *Pna2₁*, and the symmetry elements halved from eight ($E$, $C_2$, $2C_2$, $i$, $\sigma_h$, and $2\sigma_v$) to four ($E$, $C_2$, and $2\sigma_v$), which means the DMAGeI₃ is a *mmm*F*mm*2-type ferroelectric with two equivalent polarization directions[42].

Second harmonic generation (SHG) experiment on powder sample was used to characterize its paraelectric-ferroelectric phase transition (Supplementary Note 2). As shown in Fig. 2a, DMAGeI₃ shows a strong SHG response at room temperature, which confirms its noncentrosymmetric LTP. With increasing temperature, the SHG signal intensity gradually decreases to zero above $T_C$, indicating that the HTP structure is transformed into a centrosymmetric one. Moreover, the SHG measurement on a single crystal surface of DMAGeI₃ can further reveal the direction vector of ferroelectric polarization

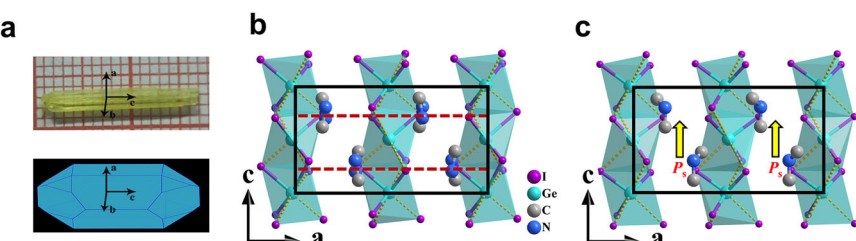

**Fig. 1 | Structural information of DMAGeI₃. a** The image and simulated crystal growth morphology of single crystal. **b, c** Packing view of the unit cell in HTP and LTP along the *b*-axis. Hydrogen ions are omitted for clarity. The red dashed lines indicate the 2-fold crystallographic rotation axes and the polarization direction are indicated by yellow arrows.

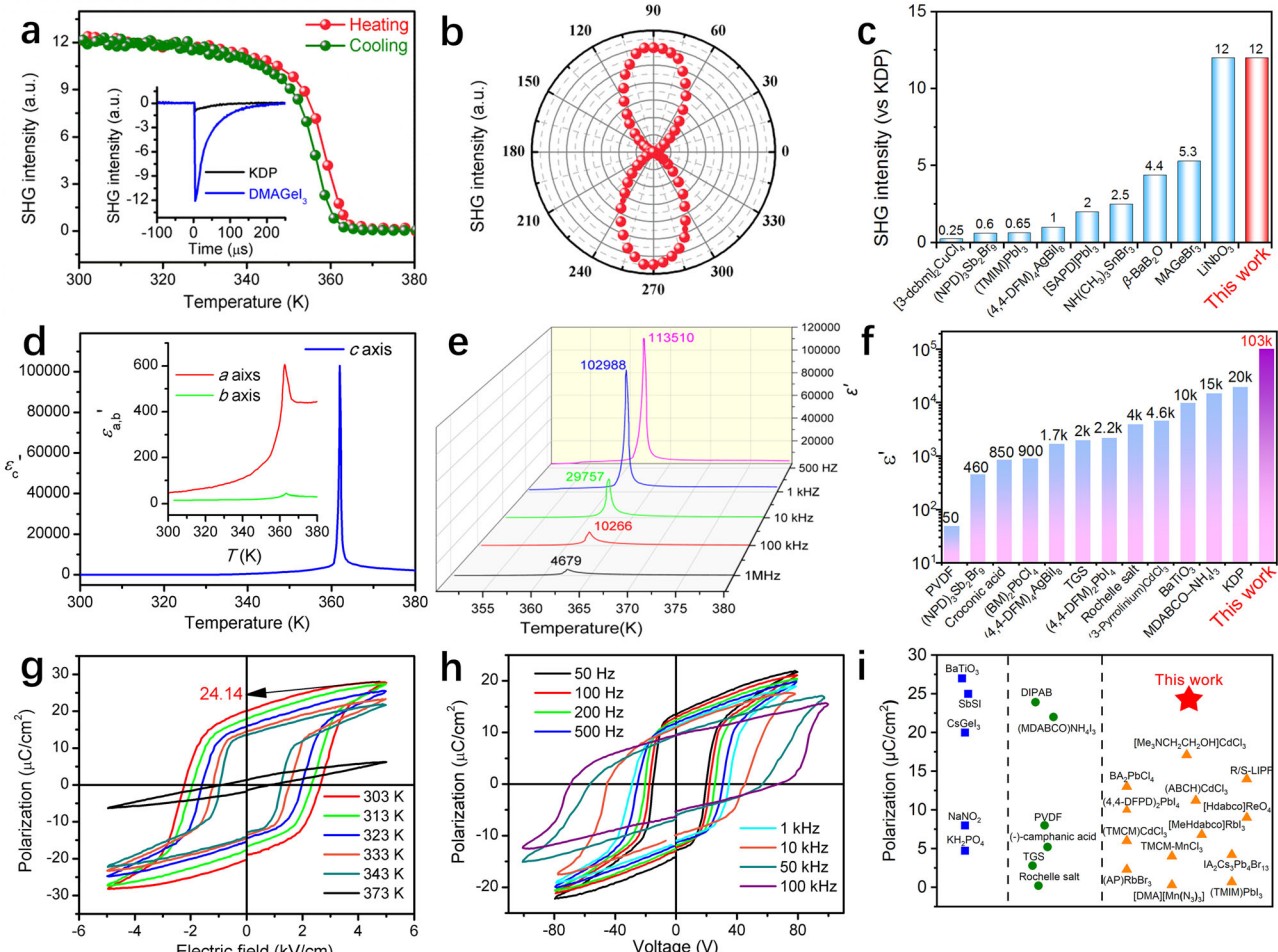

**Fig. 2 | Ferroelectric properties of DMAGeI₃. a** The SHG intensity of DMAGeI₃ as a function of temperature on powder samples. Inset: the comparison of SHG signals between DMAGeI₃ and KDP powder samples (in the same range of particle diameter 300–450 μm) at room temperature. More comparisons with various particle sizes can be found in Supplementary Fig. 8, which leads the same conclusion. **b** The SHG anisotropic polar plot of DMAGeI₃ crystal, where the $\varphi$ is the angle between the *b*-axis and the polarization of incident light. **c** Comparison of SHG intensity between DMAGeI₃ and other ferroelectrics. **d** The temperature-dependent dielectric real part ($\varepsilon'$) of DMAGeI₃ measured on single crystals along the *a*, *b*, and *c*-axis at 1 kHz. **e** The temperature-dependent dielectric real part ($\varepsilon'$) of DMAGeI₃ measured on single crystals along the *c*-axis at various frequencies. **f** Comparison of the maximum value of $\varepsilon'$ of DMAGeI₃ and other ferroelectrics. **g, h** The typical polarization-electric field (*P-E*) hysteresis loops measured along the *c*-axis at various temperatures and at different frequencies. **i** Comparison of polarization values between DMAGeI₃ and other ferroelectrics.

(Supplementary Note 2 & Supplementary Fig. 9)[43,44]. As shown in Fig. 2b, the angle-dependent SHG intensity shows a clear bipolar behavior, namely its intensity reaches the maximum (minimum) when the polarization direction of incident light is parallel (perpendicular) to the spontaneous polarization direction (i.e., *c*-axis) of DMAGeI₃ crystal. It is worth to mention that the SHG strength of DMAGeI₃ (of power sample) is more than ten times that of KDP (Fig. 2a), comparable to that of previous reported hybrid halide antiperovskite Cs₃Cl(HC₃N₃S₃) (11.4 × KDP)[45], and better than many other reported ferroelectrics (Fig. 2c and Supplementary Table 4). Thus, a direct application of DMAGeI₃ crystal is the nonlinear optical converter, which can omit bright green light via photoluminescence, as demonstrated in Supplementary Fig. 10.

Paraelectric-ferroelectric phase transition is generally accompanied by an obvious dielectric anomaly. The temperature-dependent dielectric real part ($\varepsilon'$) of DMAGeI₃ was measured on single crystals along three crystallographic axes at various frequencies. As shown in Fig. 2d, e, the dielectric constant along the *c*-axis ($\varepsilon'_c$) is much larger than those along the other two axes, especially in the region around $T_C$. This dielectric anisotropy can be ascribed to this *mmm*F*mm*2-type phase transition that only allows spontaneous polarization along the *c*-

axis. It is noteworthy that the value of $\varepsilon'_c$ reveals an obvious frequency-dependence. Specifically, as the frequency decreases from 1 MHz to 1 kHz, the peak value of $\varepsilon'_c$ at $T_C$ increases from 4679 to 102988 which is about 3269 times of its dielectric constant (~31.5) at room temperature (Fig. 2e). To our knowledge, this is the largest dielectric response in the reported hybrid perovskite materials to date (Fig. 2f & Supplementary Table 5). Additionally, according to the Curie–Weiss law, the $C_{para}$ and $C_{ferro}$ are calculated to be 9664 and 17469 K at 1 kHz, respectively. The ratio of $C_{ferro}/C_{para}$ is 1.8 (smaller than 4), which discloses the characteristics of second-order ferroelectric phase transition (Supplementary Fig. 11).

The polarization-electric field (*P-E*) hysteresis loop is the most direct evidence of ferroelectricity. Figure 2g shows the *P-E* loops at various temperatures. Above $T_C$ (e.g., 373 K), the polarization response to the applied field presents almost linear behavior with a very narrow loop, while the well-defined *P-E* hysteresis loops were observed in the range of 343–303 K (below $T_C$), as expected for the paraelectricity and ferroelectricity of the HTP and LTP. As the temperature decreases in the LTP, both the saturation polarization ($P_s$) and remnant polarization ($P_r$) increase gradually, and reach up to 24.14 μC/cm² and 20.26 μC/cm² at 303 K, respectively. Remarkably, the $P_s$ of DMAGeI₃ is larger than

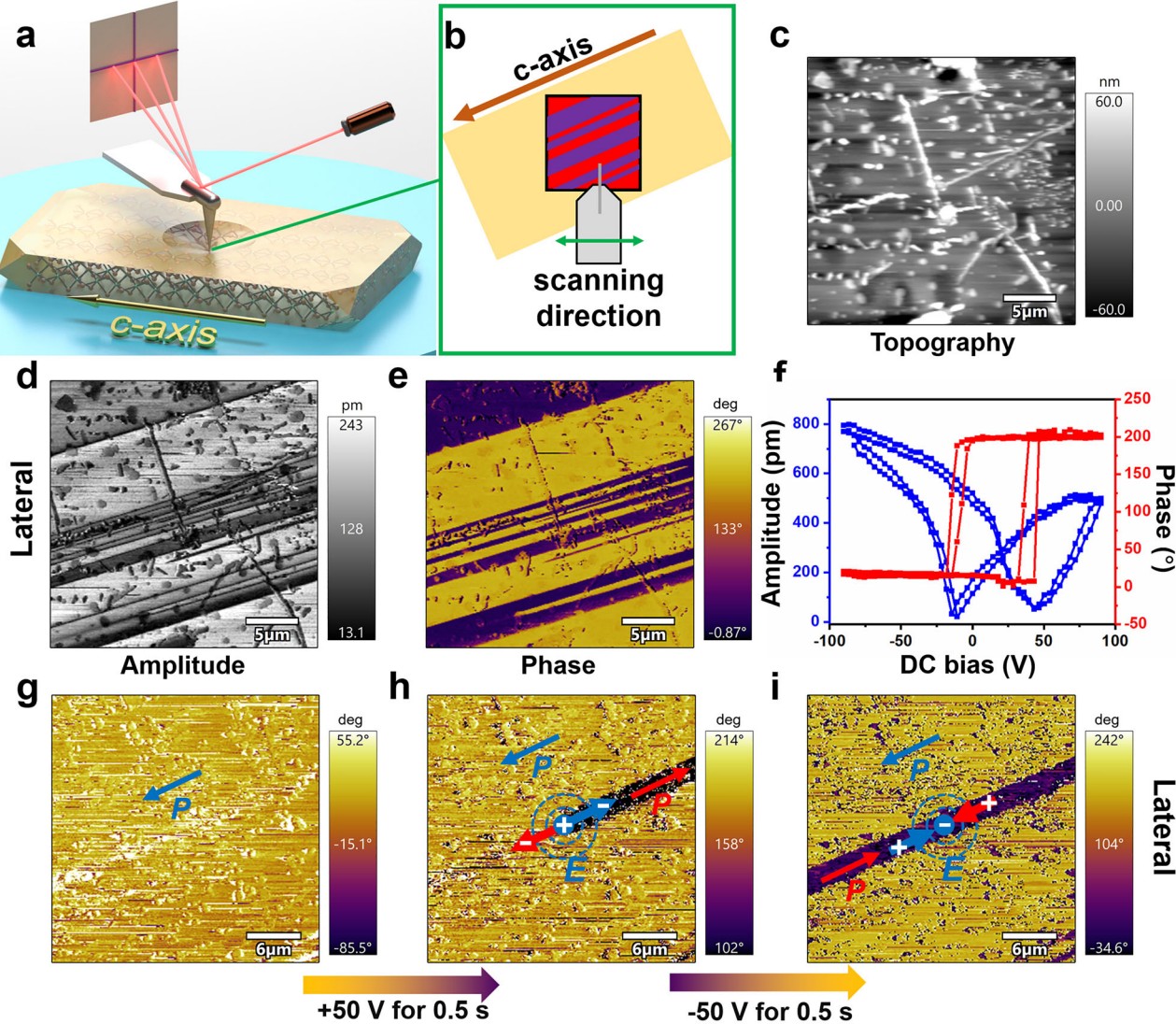

**Fig. 3 | Ferroelectric domain structures and polarization switching of DMAGeI₃.** **a**, **b** Three-dimensional and overhead schematic diagram of lateral mode of PFM for DMAGeI₃. **c**–**e** Topography, lateral amplitude, and phase images. **f** Lateral PFM amplitude (blue) and phase (red) signals as functions of the tip voltage, showing local PFM hysteresis loops. **g** Initial phase image with 30 × 30 μm². **h** Phase image after applying positive bias of +50 V for 0.5 s in the center of the area. **i** Succeeding phase image after applying negative bias of −50 V for 0.5 s at the same location.

those of most reported famous organic and OIHP ferroelectrics such as [Me₃NCH₂CH₂OH]CdCl₃ (17.1 μC/cm²)[28], [MeHdabco]RbI₃ (6.8 μC/cm²)[29], [Me₃NCH₂Cl]MnCl₃ (4.0 μC/cm²)[38], and very close to that of BaTiO₃ (Fig. 2i & Supplementary Table 6).

Another notable feature of DMAGeI₃ ferroelectricity is its small coercive field ($E_c$) values (0.8–2.2 kV/cm), which are one order of magnitude lower than that of the recently reported all-inorganic perovskite CsGeI₃ (~40 kV/cm)[41], and two orders of magnitude lower than those of PVDF (~500 kV/cm) and its copolymers[39]. More comparisons with other hybrid ferroelectrics can be found in Supplementary Table 7. Such a small $E_c$ is advantageous for energy-saving operations in devices. Furthermore, the polarization switching speed is also a key property for ferroelectric applications, thus the frequency dependence of P-E hysteresis loops is investigated, as shown in Fig. 2h. With rising frequency, the values of $P_s$ and $P_r$ slightly decrease, while the coercive voltage increases. Its ferroelectric polarization can be switched at a frequency up to 100 kHz, which shows excellent high-frequency performance for macroscopic crystals with large scale domains. Given the large $P_s$ value in combination with the small $E_c$ and the high-frequency performance, DMAGeI₃ represents promising

hybrid ferroelectric materials for device applications. Moreover, the piezoelectric response has also been measured by using the quasistatic method. At room temperature, the $d_{33}$ of the DMAGeI₃ crystal reaches 3 pC/N, along the polar axis [001] under a testing frequency of 110 Hz (Supplementary Fig. 13).

For ferroelectrics, the microscale domain structure is another essential property of polarity, which can be characterized by using the piezoresponse force microscopy (PFM) technology[46]. By adjusting the lateral (in-plane) or vertical (out-of-plane) mode in PFM, the relative strength of piezoelectric response and polarization directions from different components can be harvested. Figure 3a, b display a schematic diagram of PFM in-plane testing on DMAGeI₃ crystal along the c-axis (polarization direction), and the obtained topography of the crystal surface is shown in Fig. 3c. In the lateral mode, the observed obvious stripy domain walls parallel to the c-axis eliminate the interference of topography and confirm the existence of ferroelectric polarization (Fig. 3d). Combined with amplitude, the two sides of the domain wall show the response to different polarization directions, displaying obvious phase difference (Fig. 3e). In contrast, the piezoelectric signals do not give a component in the vertical direction

(Supplementary Fig. 14), verifying the results of crystal structure determination (*mmm*F*mm*2-type ferroelectric).

To further illustrate the ferroelectricity of DMAGeI₃, the ferroelectric domain is switched using PFM. As shown in Fig. 3f, the butterfly-shaped amplitude (blue) and phase hysteresis loops (red) provide evidence for ferroelectric switching. A single domain state region with $30 \times 30 \, \mu m^2$ is selected to demonstrate the domain inversion of DMAGeI₃ (Fig. 3g). After applying positive bias of +50 V for 0.5 s, in the sectional area parallel to the *c*-axis on the right side of the point where the bias was applied, the polarization direction is switched (Fig. 3h). On the contrary, when giving a bias of −50 V for 0.5 s, the domain in the left area is reversed, resulting in the appearance of the stripy domain (Fig. 3i). Subsequently, a higher bias is applied to the center point, and the left area turns to the same phase, which is obviously different from the reverse polarization direction on the right (Supplementary Fig. 15). Our PFM measurement results strongly prove that DMAGeI₃ possesses stable and switchable polarization.

## Discussion

To further understand the prominent ferroelectricity of DMAGeI₃, we performed density-functional-theory (DFT) calculations to reveal underlying mechanisms. Our DFT calculation leads to lattice constants close to the experimental LTP one (Supplementary Fig. 17). And its electronic structure is shown in Supplementary Fig. 18, suggesting a good insulator with a band gap 2.79 eV. The valance band maximum is contributed by Ge and I. The theoretical value of polarization is estimated as $29.09 \, \mu C/cm^2$ at the ground state (i.e., $T = 0 \, K$), which agrees well with (and slightly higher than) the experimental value at room temperature.

Then the ferroelectric origin of DMAGeI₃ is analyzed. First, the $4s^2$ lone pair of $Ge^{2+}$ ion is a strong driving force of ferroelectricity, as appeared in the all-inorganic 3D perovskite CsGeI₃[41]. As illustrated in Fig. 4a, the $4s^2$ lone pairs form a zigzag pattern along the inorganic chain, leading to an uncompensated dipole along the *c*-axis in the LTP. In contrast, in the HTP structure, the local dipoles from $4s^2$ lone pairs are fully compensated, as compared in Fig. 4b. Second, besides the contribution from $Ge^{2+}$'s $4s^2$ lone pair, the effect of DMA cation is also non-negligible. Both the bodily movement of this +1 charged group and stereo orientation of this asymmetric structure will generate a local dipole. A simple estimation of their individual contributions are $16.18 \, \mu C/cm^2$ (from the DMA cations) and $13.62 \, \mu C/cm^2$ (from the GeI₃ framework) respectively (see Supplementary Note 3 and Supplementary Fig. 19). In other words, the large polarization of DMAGeI₃ is due to the synergistic effect of inorganic and organic parts, with almost half-half weights.

The ferroelectric switching process of DMAGeI₃ can be also simulated by rotating the DMA cations, as depicted in Fig. 4a. If the four DMA cations in one unit cell rotate synchronously, the switching barrier energy is 292.9 meV, which is not as low as expected. However, this barrier can be significantly reduced by rotating the DMA cations one by one, as compared in Fig. 4b, where the first and second barriers of the rotating steps are 109.6 meV and 115.6 meV, respectively. For comparison, the DFT switching barrier for a BaTiO₃ unit cell is ~8.3 meV and its experimental coercive field $E_c$ is <10 kV/cm[12]. But it should be noted that the unit-cell volume of DMAGeI₃ is ~16.5 times of BaTiO₃. Thus, the two one-by-one barriers are only ~6.6 meV and ~7.0 meV for DMAGeI₃ when normalized to the volume of BaTiO₃. Compared with BaTiO₃, the proximate $P_s$ and lower barrier of DMAGeI₃ suggest an even smaller $E_c$, as observed in our experiments.

Furthermore, its prominent SHG signal can be also well reproduced theoretically. The nonlinear optical susceptibility tensors ($d_{ij}$'s) of LTP of DMAGeI₃ are calculated, which determine the intensity and anisotropy of SHG signals. For DMAGeI₃ with space group $Pna2_1$, its SHG intensity on the *bc* crystalline plane can be expressed as[47]: $I_{DMAGeI_3} \propto (2d_{24} \sin 2\varphi)^2 + (2d_{32}\cos^2\varphi + 2d_{33}\sin^2\varphi)^2$, where $\varphi$ is the angle between the polarization direction of incident light and *b*-axis (more details can be found in Supplementary Note 4). As shown in Fig. 4c, the calculated SHG plot of DMAGeI₃ shows a bipolar behavior, and the strongest signal appears when the polarization direction of incident light is along the *c*-axis of crystal, consistent with the experimental observation.

For comparison, the $d_{ij}$'s of KDP room temperature phase (space group $I\bar{4}2d$) is also calculated, which agrees well with the experimental values (Supplementary Fig. 20). As expected, these values are systematically smaller than those of DMAGeI₃. Thus the SHG signal of KDP is much smaller than that of DMAGeI₃, since generally the SHG response is proportional to the square of $d_{ij}$'s, as compared in Fig. 4c.

Our study unambiguously demonstrated the excellent comprehensive ferroelectric properties of DMAGeI₃ with quasi-one-dimensional structure, which is a breakthrough of organic-inorganic hybrid ferroelectrics as well as low-dimensional ferroelectrics. Its remarkable ferroelectricity originates from the synergistic overlap of two robust mechanisms: ordering of polar cations and soft mode of $4s^2$ lone pairs, which doubles its net polarization and keeps its coercivity at a low level. This successful example provides a reliable and simple route to pursue more superior ferroelectrics or even multiferroics in organic–inorganic

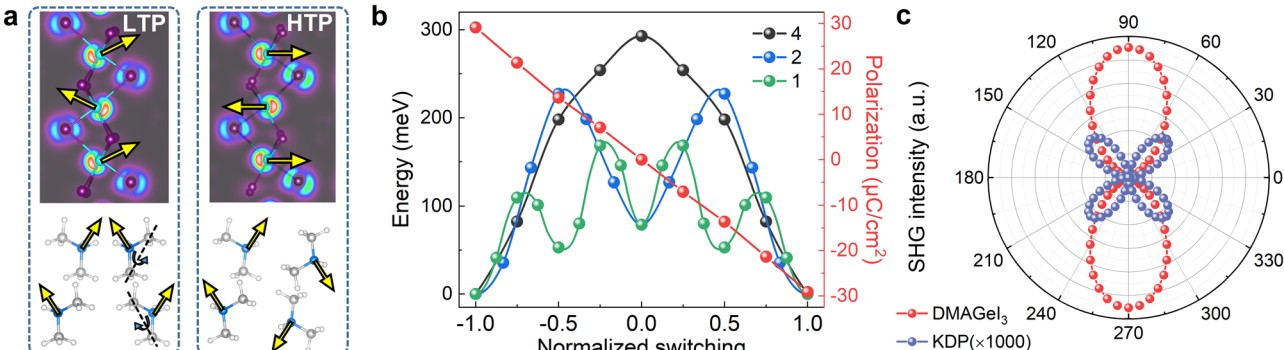

**Fig. 4 | Ferroelectricity revealed by ab initio calculations. a** The schematic polarization of the GeI₃ framework (upper) and DMA cations (lower). The yellow arrows correspond to the local dipoles' directions. The lone pairs of Ge's $4s^2$ electron clouds can be clearly visualized as a driving force of dipoles. And the dipoles of DMA cations can be flipped by rotation. **b** The calculated energy barriers and evolution of polarization of ferroelectric switching. The barrier depends on the switching process, namely how many DMA cations (1, 2, or 4) rotate synchronously in a unit cell. The one-by-one rotation of DMA rotation leads to a lower barrier. Note that the theoretical barrier only defines the upper limit, while the real barrier can be even lower, especially at room temperature. **c** The comparison of calculated SHG plots of DMAGeI₃ and KDP crystals (both on the *bc* plane). The 0° (90°) is along the *b*-axis (*c*-axis).

hybrid systems, to finally reach the aim of commercial applications. The opportunities are vast for emerging hybrid germanium halide perovskites, and the ones equipped with multipolar characteristics and outstanding piezoelectric performance may be available in the near future.

## Methods

### DSC, SHG, and XRD measurements

Differential scanning calorimetry (DSC) measurements were carried out by using a NETZSCH DSC 200F3 instrument under the nitrogen atmosphere, where the dry powder of $DMAGeI_3$ (28.9 mg) were heated and cooled with a rate of 10 K/min in the temperature ranges of 293–390 K. SHG switching experiments were measured with powder samples by pulsed Nd: YAG (1064 nm, Vibrant 355 II, OPOTEK), and the temperature varies from 300 K to 380 K. The powdered $KH_2PO_4$ (KDP) was used as the reference. The SHG phase matching tests were performed with a particle size range of 62–375 µm at room temperature. Furthermore, the SHG intensity anisotropy was investigated by using Witec alpha 300 on single-crystal surfaces. Variable-temperature single-crystal X-ray diffraction data were collected on a Rigaku VarimaxTM DW diffractometer with Mo Kα radiation ($\lambda = 0.71073$ Å). Data processing with empirical absorption correction was conducted by using the CrysAlisPro 1.171.40.14e (Rigaku OD, 2018). The crystal structures were confirmed by direct methods and refined by full-matrix least-squares methods based on $F^2$ through the OLEX2 and SHELXTL (version 2018) software package. All non-hydrogen atoms were refined anisotropically and all hydrogen atoms were generated geometrically in suitable positions.

### Dielectric and ferroelectric measurements

For the dielectric measurements, the sample of $DMAGeI_3$ was made with single-crystals cut perpendicular to the *a*-, *b*-, and *c*-axis respectively. Silver conduction paste deposited on the plate surfaces was used as the electrodes. The temperature-dependent dielectric constants were performed on the Tonghui TH2828A instrument under the frequency range from 1 kHz to 1 MHz with an applied electric field of 1 V. The single crystal electrodes of $DMAGeI_3$ were prepared for the *P-E* hysteresis loop measurements by Sawyer-Tower method. The ferroelectric polarization imaging and local switching studies on the bulk crystal surface were carried out by using a resonant-enhanced PFM (MFP-3D, Asylum Research).

### PXRD and TGA measurements

Powder XRD data were measured using a Rigaku D/MAX 2000 PC X-ray diffraction system with Cu Kα radiation in the 2θ range of 5°–50° with a step size of 0.02°. The thermogravimetric analysis (TGA) were performed on the NETZSCH TG 209F3 instrument in the range of 298–1000 K with a heating rate of 20 K/min.

### DFT Calculations

The electronic structure calculations were performed based on projector augmented wave pseudopotentials, as implemented in Vienna ab initio Simulation Package (VASP)[48]. Plane-wave cutoff energy of 500 eV and $3 \times 4 \times 4$ *k*-point meshes were used. The lattice constants and the atomic positions were fully relaxed with the convergence criteria of $10^{-5}$ eV and 0.01 V/Å for energy and force, respectively. The exchange-correlation potential was approximated in the form of Perdew–Burke–Ernzerhof (PBE) functional and the van der Waals (vdW) interaction was considered by adding D3 Grimme correction[49,50]. Other choices of functional and vdW correction were also tested (Supplementary Fig. 17). The climbing nudged elastic band method was used to determine the paths and energy barrier of ferroelectric switching[51]. Ferroelectric polarization was calculated by using the Berry phase method[52]. The second-harmonic nonlinear optical susceptibility tensors for SHG were calculated using the exciting package[53].

## Data availability

The experimental cif files can be found in CCDC ($DMAGeI_3$: 2227301 (223 K), 2227302 (253 K), 2213657 (293 K), 2220591 (373 K); $DMAGeCl_3$: 2213654 (293 K); $DMAGeBr_3$: 2213655 (293 K)). Source data is provided with this paper. Source data are provided with this paper.

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

## Acknowledgements

This work is supported by the National Natural Science Foundation of China (Grant Nos. 21991141, 92056112, 12274069, 11834002). We thank the Big Data Computing Center of Southeast University for computational resource.

## Author contributions

D.-W.F.,Y.Z., S.D., and Z.-X.Z. conceived the project. D.W.F. and Y.Z. designed the experiments. S.D. proposed the theoretical mechanisms. K.D. prepared the samples and performed the DSC, single crystal measurement and analysis. Z.-X.Z. contributed to dielectric and P-E loops measurements and analysis. H.Y. performed the DFT calculations guided by S.D. C.S. contributed to the analysis of PFM. Y.A.X., D.W., G.D. and Y.M.Y. contributed to PFM and SHG measurements. Z.-X.Z., D.W.F., S.D. and Y.Z. wrote the manuscript, with inputs from all other authors.

## Competing interests

The authors declare no competing interests.
