## [Peer Review File · Nature Communications]

Superior ferroelectricity and nonlinear optical response in a hybrid germanium iodide hexagonal perovskiteREVIEWER COMMENTS

Reviewer #1 (Remarks to the Author):

Ding et. al, reported a quasi-one-dimensional hybrid perovskite ferroelectric DMAGel3 with a large P_s ($24.14 \mu\text{C}/\text{cm}^2$), low E_c (below $2.2 \text{ kV}/\text{cm}$) and strong SHG intensity (12 times of KH_2PO_4). Calculation is carried out to understand its ferroelectric performance. Although the technical aspect of the work is fine, conceptual breakthrough is insufficient. Hybrid perovskite ferroelectrics are expected to have large P_s , high T_c and multiple polar axes features for functional applications in photoelectric devices, micromechanical sensors or energy-storage devices. Although the authors have mentioned many times its promising device applications (line 171, 178 etc.), the work only focuses on characterization. A suitable application should be demonstrated for such a high ferroelectric material. The following points need to be addressed before it can be further considered:

1. There is an earlier reported 1D $[\text{Me}_3\text{NCH}_2\text{CH}_2\text{OH}]\text{CdCl}_3$ hybrid perovskite ferroelectric with high P_s of $17.1 \mu\text{C}/\text{cm}^2$ for mechanical energy harvesting applications (Chem. Mater. 2020, 32, 8333–8341). The author seems to overlook this paper. Pls cite it and make a comparison. Generally speaking, as the symmetry is lowered from 3D to 1D, it is easier to obtain ferroelectricity. Therefore, demonstration on a 2D analog is better.
2. Seeking hybrid perovskite ferroelectric semiconductor with superior photoelectric performance is one goal in this research area. How about the bandgap, semiconducting properties and mechanical property of DMAGel3. Can its superior ferroelectricity be used for any device application?
3. It is a common thing for ferroelectric materials show SHG signal below curie temperature due to its centrosymmetry breaking. To verify its superior nonlinear optical response, the author claim “the SHG strength of DMAGel3 (of powder sample) is more than ten times that of KH_2PO_4 ”. However, to quantification SHG susceptibilities, SHG coefficient $\chi(2)$ is commonly used to evaluate the performance of NLO materials (J. Am. Chem. Soc. 2021, 143, 16095–16104). In an earlier reported hybrid Germanium iodide perovskite paper (J. Am. Chem. Soc. 2015, 137, 6804–6819), $\chi(2)$ is used to confirm the strongest NLO. Can author estimate the SHG coefficient and make a comparison with the mentioned two papers?
4. Can author provide information on the stability of the crystal?
5. Authors give a beautiful polarization switching based on PFM. However, the surface of material does not look clean or smooth enough. Does this will affect the performance of devices?

6. In Fig. 4c, compared with DMAGel3's bipolar, KPD seems show multi-polar behavior. Is it possible to change the crystal's polar or ferroelectric domains by tuning molecular composition ratios ?

7. Pay attention to some language mistakes in the manuscript. In line 56, Ps's should be Ps; In line 261, "Thus the SHG signal of KDP of ...", there is one extra "of".

Reviewer #2 (Remarks to the Author):

Kun Ding et al. reported a hybrid germanium iodide hexagonal perovskite DMAGel3 with a large Ps (24.14 $\mu\text{C}/\text{cm}^2$), a low Ec (2.2 kV/cm), and the strongest SHG (12 times of KDP) intensity in OIHP family. The theoretical simulation revealed large Ps and SHG response originate from the synergistic effects of the stereochemically active lone pair of Ge²⁺ and the ordering of organic cations. Moreover, the low kinetic energy barrier of small DMA cation leads to low Ec. Considering that this work presents a major advance in Ps and SHG response of OIHP ferroelectrics, I think this manuscript is suitable for publication in Nature Communications after minor revision. However, the manuscript also has some issues that need to be solved.

(1) Figure 2e shows the obvious frequency dependence of the temperature-dependent dielectric real part (ϵ') in the c-axis direction (polar axis direction). For the completeness of experimental data and the anisotropic analysis, it is suggested that the authors provide the dielectric curves at different frequencies in the directions of other two axes.

(2) The SHG intensity of powder polycrystalline sample is closely related to the particle size. The particle sizes of DMAGel3 polycrystalline powder and KDP sample used to compare SHG strength should be stated.

(3) Though adding (deducting) electrons in the pure DMA group (the framework) gives a close result to global polarization, the structures are seriously modified. The individual contribution to polarization may be different from the authors' results. Is there any other method to confirm the individual polarization?

(4) There is a certain relationship between the piezoelectric response and polarization of ferroelectrics. Given such a large Ps (24.14 $\mu\text{C}/\text{cm}^2$) of DMAGel3, I suggest the authors to further measure its piezoelectric performance.

(5) The SHG experiment is described in Supplementary Note 2. Please correct it in the manuscript.

Reviewer #3 (Remarks to the Author):

The authors reported a quasi-one-dimensional perovskite DMAGeI₃, which show excellent comprehensive ferroelectric performances comparable to commercial inorganic ferroelectric perovskites. Interestingly, the cooperation of ordering of polar cations and soft mode of 4s² lone pairs in this perovskite results in large P_s of 24.14 μC/cm², low E_c of 2.2 kV/cm, as well as strong SHG response. This new structural perspective would provide a feasible path to more superior ferroelectrics. Experimental characterizations and theoretical calculations are systematic and reliable. I think this work presents an exciting and significant advancement in ferroelectrics and this manuscript is suitable for publishing in Nature Communications before the following issues are addressed.

1. The reversibility of physical characteristics is closely related to thermal stability. How about the thermal stability (i.e. melting point) of the DMAGeI₃ crystal?
2. The SHG is performed in the ferroelectric phase; therefore, it is necessary to confirm the phase purity.
3. What about the SHG intensity vs the particle size of crystals? The SHG responses of the tested material and KDP should be tested at the same particle sizes. Relevant phase matching tests are necessary.
4. The citation formats for some figures are inconsistent, such as “Supplementary Fig. 8”, and “Supplementary Fig. S12”.
5. The spontaneous polarization for DMAGeI₃ is resulting from the ordering of organic cations. According to the Curie-Weiss law, the type of ferroelectric phase transition needs to be further elucidated from the temperature-dependent dielectric curves.
6. The authors mentioned the synthesis and structures of other halides in the supporting information. It is suggested to provide detailed experimental and conclusion information about this series of compounds.
7. The authors are suggested to cite the related references on hybrid halide perovskites, such as “A Hybrid Antiperovskite with Strong Linear and Second-Order Nonlinear Optical Responses, *Angew. Chem. Int. Ed.*, 2022, e202211151” and “A Hybrid Halide Perovskite Birefringent Crystal, *Angew. Chem. Int. Ed.*, 2022, 61, e202202746”.

Response to Reviewer #1

General Comment: Ding et al, reported a quasi-one-dimensional hybrid perovskite ferroelectric DMAGeI₃ with a large P_s (24.14 $\mu\text{C}/\text{cm}^2$), low E_c (below 2.2 kV/cm), and strong SHG intensity (12 times of KH₂PO₄). Calculation is carried out to understand its ferroelectric performance. Although the technical aspect of the work is fine, the conceptual breakthrough is insufficient. Hybrid perovskite ferroelectrics are expected to have large P_s , high T_c and multiple polar axes features for functional applications in photoelectric devices, micromechanical sensors or energy-storage devices. Although the authors have mentioned many times its promising device applications (line 171,178 etc.), the work only focuses on characterization. A suitable application should be demonstrated for such a high ferroelectric material. The following points need to be addressed before it can be further considered:

Response: We thank the reviewer for his/her carefully reviewing. Following this suggestion, an example of applications as non-linear optical converter has been demonstrated, as shown in Response to Comment 2. Of course, we admit that the present work is a physical+chemical investigation, mainly orientated to the properties characterization of a new crystal. Certainly, more applications need following works from multidisciplinary researchers. Thus, we have toned down the claim on device applications, but focus more on the material properties. We have addressed all the concerns of the reviewer and made proper modifications to our manuscript.

Comment 1: There is an earlier reported 1D [Me₃NCH₂CH₂OH]CdCl₃ hybrid perovskite ferroelectric with high P_s of 17.1 $\mu\text{C}/\text{cm}^2$ for mechanical energy harvesting applications (Chem. Mater., 2020, 32, 8333-8341). The author seems to overlook this paper. Pls cite it and make a comparison. Generally speaking, as the symmetry is lowered from 3D to 1D, it is easier to obtain ferroelectricity. Therefore, demonstration on a 2D analog is better.

Response: Thanks for bring this work to our attention. We have carefully studied and cited this paper (Chem. Mater. 2020, 32, 8333-8341) as reference 28. Furthermore, the comparison with [Me₃NCH₂CH₂OH]CdCl₃ are shown in Fig. 2i and Supplementary Table 6.

As the reviewer referred, it is easier to obtain ferroelectricity when the symmetry is lowered from 3D to 1D. However, at least to our best knowledge, there are no 2D Ge-based ferroelectric hybrid materials reported so far^{R1-R10}. In future work, we will devote more efforts to exploring 2D Ge-based ferroelectric materials.

The corresponding changes in the revised version:

Remarkably, the P_s of DMAGeI₃ is larger than those of most reported famous organic

and OIHP ferroelectrics such as $[\text{Me}_3\text{NCH}_2\text{CH}_2\text{OH}]\text{CdCl}_3$ ($17.1 \mu\text{C}/\text{cm}^2$)²⁸, $[\text{MeHdabco}]\text{RbI}_3$ ($6.8 \mu\text{C}/\text{cm}^2$)²⁹, $[\text{Me}_3\text{NCH}_2\text{Cl}]\text{MnCl}_3$ ($4.0 \mu\text{C}/\text{cm}^2$)³⁸, and very close to that of BaTiO_3 (Fig. 2i & Supplementary Table 6).

Fig. 2 | i, Comparison of polarization values between DMAGeI_3 and other ferroelectrics.

Supplementary Table 6 | Comparison of spontaneous polarization P_s in a variety of materials and DMAGeI_3 in this work.

Compound	P_s ($\mu\text{C}/\text{cm}^2$)	Reference
TGS	3.8	S41
BaTiO_3	27	S42
PVDF	8	S42
CsGeI_3	20	S43
Croconic acid	20	S42
$\text{MDABCO-NH}_4\text{I}_3$	22	S35
TMCM-MnCl_3	4	S40
TMCM-CdCl_3	6	S40
QP	4.5	S44
Rochelle salt	0.25	S42
$[\text{NH}_4][\text{Zn}(\text{HCOO})_3]$	0.68	S45
$(3\text{-bromopropylammonium})_2\text{PbBr}_4$	4.8	S46
$(4,4\text{-DFPD})_2\text{PbI}_4$	10	S37
$[\text{CH}_3(\text{CH}_2)_3\text{NH}_3]_2(\text{CH}_3\text{NH}_3)\text{Pb}_2\text{Br}_7$	3.6	S21
$(\text{C}_4\text{H}_9\text{NH}_3)_2(\text{C}_2\text{H}_5\text{NH}_3)_2\text{Pb}_3\text{Br}_{10}$	5	S47
$(\text{C}_4\text{H}_9\text{NH}_3)_2(\text{CH}_3\text{NH}_3)_2\text{Pb}_3\text{Br}_{10}$	2.9	S48
$(\text{EATMP})\text{PbBr}_4$	0.95	S49
$[\text{Hdabco}]\text{BF}_4$	4.9	S50
$[\text{Hdabco}]\text{ClO}_4$	4	S51
diisopropylammonium bromide	23	S52
$[\text{Me}_3\text{NCH}_2\text{CH}_2\text{OH}]\text{CdCl}_3$	17.1	S53

DMAGeI ₃	24.14	This work
-------	-----------

References

- [R1] Stoumpos C. C. *et al.* Hybrid germanium iodide perovskite semiconductors: Active lone pairs, structural distortions, direct and indirect energy gaps, and strong nonlinear optical properties. *J. Am. Chem. Soc.* **137**, 6804–6819 (2015).
- [R2] Zhang, Y. *et al.* Ferroelectricity in a semiconducting all-inorganic halide perovskite. *Sci. Adv.* **8**, eabj5881 (2022).
- [R3] Liu, Y. *et al.* Hybrid germanium bromide perovskites with tunable second harmonic generation. *Angew. Chem. Int. Ed.* **61**, e202208875 (2022).
- [R4] Zhao, X. M. *et al.* Polar molecule-based material with optic–electric switching constructed by polar anions. *Inorg. Chem.* **59**, 5475–5482 (2020).
- [R5] Li, X. Y. *et al.* Stereochemically active lone pairs and nonlinear optical properties of two-dimensional multilayered tin and germanium iodide perovskites. *J. Am. Chem. Soc.* **144**, 18030–18042 (2022).
- [R6] Cheng, P. F. *et al.* Lead-free, two-dimensional mixed germanium and tin perovskites. *J. Phys. Chem. Lett.* **9**, 2518–2522 (2018).
- [R7] Cheng, P. F. *et al.* (C₆H₅C₂H₄NH₃)₂GeI₄: A layered two-dimensional perovskite with potential for photovoltaic applications. *J. Phys. Chem. Lett.* **8**, 4402–4406 (2017).
- [R8] Huang, L. Y., Lambrecht, W. R. L. Vibrational spectra and nonlinear optical coefficients of rhombohedral CsGeX₃ halide compounds with X= I, Br, Cl. *Phys. Rev. B*, **94**, 115202 (2016).
- [R9] Tang, L. C. *et al.* New infrared nonlinear optical crystal CsGeBr₃: Synthesis, structure and powder second-harmonic generation properties. *J. Phys. Condens. Matte.* **17**, 7275–7286 (2005).
- [R10] Chen, C. C. *et al.* “Breathing” organic cation to stabilize multiple structures in low-dimensional Ge-, Sn-, and Pb-based hybrid iodide perovskites. *Inorg. Chem. Front.* **9**, 4892–4898 (2022).

Comment 2: Seeking hybrid perovskite ferroelectric semiconductor with superior photoelectric performance is one goal in this research area. How about the bandgap, semiconducting properties and mechanical property of DMAGeI₃. Can its superior ferroelectricity be used for any device application?

Response: We thank the reviewer for this instructive suggestion. The semiconducting properties of the DMAGeI_3 are shown in Supplementary Fig. 16. The calculated maximum of valence and minimum of conduction are coherently located at the same point (Γ), exhibiting a direct band gap ~ 2.8 eV, advantageous for visible light photoelectric functions. The bands near Fermi level are mainly contributed by the GeI_3 framework. Its mechanical properties are not closely related to the present work, but will be the next step in our plan.

The polar crystal structure of DMAGeI_3 with high spontaneous polarization endows it with strong SHG strength, making it attractive in optical-related applications. For example, the bright green light can be clearly observed on the crystal of DMAGeI_3 under 1064 nm linearly polarized light, which indicates that the sample generates strong second-harmonic light with wavelength of 532 nm (Supplementary Fig. 9a). This demonstrates that DMAGeI_3 has potential applications in the non-linear optical converter (Supplementary Fig. 9b).

Supplementary Fig. 9 | **a**, Experimental diagram and photograph of crystal. **b**, Schematic of the non-linear optical converter.

Comment 3: It is a common thing for ferroelectric materials show SHG signal below curie temperature due to its centrosymmetry breaking. To verify its superior nonlinear optical response, the author claim "the SHG strength of DMAGeI_3 (of powder sample) is more than ten times that of KH_2PO_4 ". However, to quantify SHG susceptibilities, SHG coefficient $\chi(2)$ is commonly used to evaluate the performance of NLO materials (J. Am. Chem. Soc. 2021, 143, 16095-16104). In an earlier reported hybrid Germanium iodide perovskite paper (J. Am. Chem. Soc. 2015, 137, 6804-6819), $\chi(2)$ is used to confirm the strongest NLO. Can the author estimate the SHG coefficient and make a comparison with the mentioned two papers?

Response: Thanks for bringing these two papers to our attention. In fact, we have reported the calculated SHG coefficients of DMAGeI₃ in the manuscript. Both $\chi_{ij}^{(2)}$ and d_{ij} can be used as SHG coefficients with a direct transformation $\chi_{ij}^{(2)} = 2d_{ij}^{R11, R12}$. Then the effective $\chi^{(2)}$ for a powder sample is an overall integration of d matrix (see Supplementary Note 4 for more details), which is in proportional to the square root of SHG signal I . Thus, the ratio of $\chi^{(2)}$ values of two different samples follows the formula:

$$\frac{\chi_1^{(2)}}{\chi_2^{(2)}} = \left| \frac{I_1(2w)}{I_2(2w)} \right|^{\frac{1}{2}} \quad (1)$$

In the paper (J. Am. Chem. Soc. 2015, 137, 6804–6819), a standard material AgGaS is used to compare $\chi^{(2)}$. They estimated the SHG coefficients of the samples; CsGeI₃, $\chi^{(2)} = (125.3 \pm 10.5)$ pm/V; MAgGeI₃, $\chi^{(2)} = (161.0 \pm 14.5)$ pm/V; FOGeI₃, $\chi^{(2)} = (143.0 \pm 13.5)$ pm/V and MFOGeI₃, $\chi^{(2)} = (57.2 \pm 5.5)$ pm/V, respectively. However, it should be noted that these values of $\chi^{(2)}$ were only obtained with 1800 nm incident light. In our experiment, the SHG signal was measured with 1064 nm incident light. For comparison, the testing conditions need to remain the same, namely the SHG signal should be compared at the same wavelength of incident light. From the Figure 5f in the paper (J. Am. Chem. Soc. 2015, 137, 6804–6819), the SHG coefficients $\chi^{(2)}$ of CsGeI₃, FOGeI₃ and MFOGeI₃ at the 1000 nm incident light can be estimated to be 3.78 ± 0.32 , 9.77 ± 0.92 , 20.89 ± 2.0 pm/V respectively (Figure R1 and Table R1). And at 1100 nm incident light, the SHG coefficients $\chi^{(2)}$ of CsGeI₃, MAgGeI₃, FOGeI₃ and MFOGeI₃ at the 1100 nm incident light can be estimated to be 7.32 ± 0.61 , 14.40 ± 1.30 , 36.92 ± 3.49 , 30.45 ± 2.93 pm/V respectively (Figure R1 and Table R1). Given the increasing trend of SHG intensity with increasing wavelength in the range of 1000-1800 nm, their $\chi^{(2)}$ under the 1064 nm incident light should be between the two sets of values mentioned above. For (J. Am. Chem. Soc. 2021, 143, 16095–16104), we cannot obtain effective data on the SHG coefficient under the incident light near 1064 nm, based on the existing information provided in the paper (Figure R2).

In our work, we used KDP as the standard material whose dominant d_{36} is 0.38 pm/V (i.e. $\chi_{36}^{(2)} = 0.76$ pm/V). First, our calculated SHG coefficients of both KDP and CsGeI₃ agree well with the experimental measurements (Table R1)^{R1, R13}, providing a reliable base for the comparison. Second, under the 1064 nm incident light, the dominating SHG coefficient d_{33} of DMAGeI₃ is calculated to be 23.53 pm/V, and its $\chi^{(2)}$ is 47.06 pm/V, which is larger than those of CsGeI₃, MAgGeI₃, FOGeI₃ and MFOGeI₃ under this condition. Of course, the above calculation results inevitably have certain deviation, but it is also sufficient to prove the strong SHG response of DMAGeI₃, which is better than many other reported ferroelectric hybrid

perovskites (Fig. 2c and Supplementary Table 4).

Figure R1. The figure 5f in the paper (J. Am. Chem. Soc. 2015, 137, 6804–6819). Noting the wavelength here (x -axis) is half of the incident light, i.e., the 500 nm corresponding to the 1000 nm incident light.

Figure R2. The figure 3a in the paper (J. Am. Chem. Soc. 2021, 143, 16095–16104).

Table R1 | Comparison of $\chi^{(2)}$ in a variety of materials and DMAGel₃ in this work.

Compound	$\chi^{(2)}$ (pm/V)	Reference
KDP	0.76 (1064 nm)	S58
KDP	0.80 (1064 nm)	Our calculated value
CsGeI ₃	125.3 ± 10.5 (1800 nm)	S27
CsGeI ₃	135.79 (1800 nm)	Our calculated value
CsGeI ₃	3.78 ± 0.32 (1000 nm)	S27
FOGeI ₃	9.77 ± 0.92 (1000 nm)	S27
MFOGeI ₃	20.89 ± 2.0 (1000 nm)	S27
CsGeI ₃	7.32 ± 0.61 (1100 nm)	S27
MAGeI ₃	14.40 ± 1.30 (1100 nm)	S27
FOGeI ₃	36.92 ± 3.49 (1100 nm)	S27
MFOGeI ₃	30.45 ± 2.93 (1100 nm)	S27
DMAGeI ₃	47.06 (1064 nm)	Our calculated value

References

[R11] Sutherland R. L. Handbook of nonlinear optics. CRC Press (2003).

[R12] Yao L. *et al.* Strong second- and third-harmonic generation in 1D chiral hybrid bismuth halides. *J. Am. Chem. Soc.*, **143**, 16095–16104 (2021).

[R13] Eckardt R. C. *et al.* Byer: Absolute and relative nonlinear optical coefficients of KDP, KD*P, BaB₂O₄, LiIO₃, MgO: LiNbO₃, and KTP measured by phase-matched second-harmonic generation. *IEEE J. Quant. Electr.*, **26**, 922-933 (1990).

Comment 4: Can the author provide information on the stability of the crystal?

Response: Thank the reviewer for the instructive comment. As suggested by the reviewer, we have measured the environmental and thermal stability of the crystal, which is added in the revised supporting information (Supplementary Fig. 1 & Fig. 2). First, all powder XRD (PXRD) patterns match well with the simulated one, indicating that the crystal structure of DMAGeI₃ remains unchanged even after exposing to dry air for 30 days. Second, the TGA measurement shows that DMAGeI₃ is quite stable before 400 K, which shows good thermal stability.

The corresponding changes in the revised version:

Supplementary Fig. 1 | Powder X-ray diffraction patterns of DMAGeI₃ under different test conditions.

Supplementary Fig. 2 | TGA curve of DMAGeI₃.

Comment 5: Authors give a beautiful polarization switching based on PFM. However, the surface of material does not look clean or smooth enough. Does this will affect the performance of devices?

Response: Thanks for the reviewer's professional comments. The crystal used in the PFM test were naturally grown by solution method, and its surface does not look clean or smooth enough due to the presence of micro/nano sized crystal grains. Even so, beautiful polarization switching with the domain reversal can still be clearly detected. The PFM phase and amplitude imaging show that the ferroelectric domains have a very clear 180° contrast in phase and the domains are separated by the clear domain walls (Figs. 3d, 3e), which are not directly related to the morphology. Furthermore, a region with a single domain initial state was chosen for the polarization switching test by applying bias via tip. The emerging domains

are flipped under the influence of the electric field. All these prove that in our measurements the imperfect surface has negligible effect on polarization switching.

We appreciate the reviewer's concern. In our future studies, we will try to optimize the surface of crystal to evaluate its impact on the performance of ferroelectric devices.

Comment 6: In Fig. 4c, compared with DMAGeI₃'s bipolar, KDP seems show multi-polar behavior. Is it possible to change the crystal's polar or ferroelectric domains by tuning molecular composition ratios.

Response: Thanks for this professional comment. According to the structural analysis, the symmetry breaking of DMAGeI₃ occurs with an Aizu notation of *mmmFmm2*, implying a uniaxial nature with two equivalent polarization directions. As the reviewer referred, by tuning the organic cations, it is possible to change the crystal's polarity or ferroelectric domains. For example, the uniaxial ferroelectric compound (3-pyrrolinium)-MnCl₃ can be transformed into multi-axial ferroelectric by replacing the 3-pyrrolinium cation with the Me₃NCH₂Cl cation^{R14, R15}. In 2020, Xiong *et al.* obtained a multi-axial molecular ferroelectric [3.2.1-dabco]BF₄ (3.2.1-dabco = 1,5-diazabicyclo[3.2.1]-octonium) based on the molecular ferroelectric [2.2.2-dabco]BF₄ (2.2.2-dabco = 1,4-diazabicyclo[2.2.2]-octonium) by tuning the organic cation^{R16}. Based on these excellent works, we are trying to build novel Ge-based molecular ferroelectric through such approaches as metal doping and modulation of cation, but certainly it is beyond the present work.

References

[R14] Ye H.-Y. *et al.* High-temperature ferroelectricity and photoluminescence in a hybrid organic–inorganic compound: (3-Pyrrolinium)MnCl₃. *J. Am. Chem. Soc.*, **137**, 13148–13154 (2015).

[R15] You Y.-M. *et al.* An organic-inorganic perovskite ferroelectric with large piezoelectric response. *Science*, **357**, 306–309 (2017).

[R16] Wei Z.-H. *et al.* Rational design of ceramic-like molecular ferroelectric by quasi-spherical theory. *J. Am. Chem. Soc.*, **142**, 1995–2000 (2020).

Comment 7: Pay attention to some language mistakes in the manuscript. In line 56, *P_s*'s should be *P_s*; In line 261, "Thus the SHG signal of KDP of ..." there is one extra "of".

Response: We thank the reviewer for pointing out these language mistakes. We have revised these typos and further polished the presentation, which are highlighted in blue.

Response to Reviewer #2

General Comment: Kun Ding et al. reported a hybrid germanium iodide hexagonal perovskite DMAGeI_3 with a large P_s ($24.14 \mu\text{C}/\text{cm}^2$), a low E_c ($2.2 \text{ kV}/\text{cm}$), and the strongest SHG (12 times of KDP) intensity in OIHP family. The theoretical simulation revealed large P_s and SHG response originate from the synergistic effects of the stereochemically active lone pair of Ge^{2+} and the ordering of organic cations. Moreover, the low kinetic energy barrier of small DMA cation leads to low E_c . Considering that this work presents a major advance in P_s and SHG response of OIHP ferroelectrics, I think this manuscript is suitable for publication in Nature Communications after minor revision. However, the manuscript also has some issues that need to be solved.

Response: We thank the reviewer for his/her valuable comments and recommendation.

Comment 1: Figure 2e shows the obvious frequency dependence of the temperature-dependent dielectric real part (ϵ') in the c -axis direction (polar axis direction). For the completeness of experimental data and the anisotropic analysis, it is suggested that the authors provide the dielectric curves at different frequencies in the directions of other two axes.

Response: We thank the reviewer for this instructive suggestion. For the completeness of experimental data and the anisotropic analysis, we have supplemented the dielectric curves at different frequencies along with the a - and b -axes in the revised supporting information (Supplementary Fig. 10).

The corresponding changes in the revised version:

Supplementary Fig. 10 | Temperature-dependent dielectric real part (ϵ') of DMAGeI_3 measured at various frequencies. **a, along the a -axis. **b**, along the b -axis.**

Comment 2: The SHG intensity of powder polycrystalline sample is closely related to the particle size. The particle sizes of DMAGeI_3 polycrystalline powder and KDP sample used to compare SHG strength should be stated.

Response: Thanks for this suggestion. The SHG intensity of the powder polycrystalline sample is closely related to the particle size. In the same particle size range of 300–450 μm , the DMAGeI_3 exhibits more than ten times the SHG strength of KDP (Fig. 2a). The corresponding particle sizes have been added in the caption of Fig. 2a in revised manuscript. In addition, in order to eliminate the effect of particle size on SHG strength, we also have performed the SHG phase matching test. The corresponding phase matching curve is supplemented in the revised supporting information (Supplementary Fig. 7). The SHG intensities increase with the particle sizes gradually reach saturated values.

The corresponding changes in the revised version:

Supplementary Fig. 7 | The particle size dependence of SHG intensity for both DMAGeI_3 and KDP under 1064 nm laser radiation.

Comment 3: Though adding (deducting) electrons in the pure DMA group (the framework) gives a close result to global polarization, the structures are seriously modified. The individual contribution to polarization may be different from the authors' results. Is there any other method to confirm the individual polarization?

Response: This is a good question. First, different from the reviewer's suspicion, our structures are not seriously modified when calculating the individual contribution, because no further optimization is performed after removing the DMA groups (or the GeI_3 framework)

from the DMAGeI_3 . Thus, the structure of individual DMA groups and GeI_3 framework keep the original ones as in DMAGeI_3 .

Second, as presented in Supplementary Fig. 18, the electron density difference, defined as $\Delta\rho = \rho(\text{DMAGeI}_3) - \rho(\text{DMA})$ where $\rho(\text{DMAGeI}_3)$ and $\rho(\text{DMA})$ are the electron density of DMAGeI_3 and DMA with adding hole/f.u., is calculated to support the rationality. It is obvious that the residual electrons only locate at the GeI_3 framework. Similar situation occurs for $\Delta\rho = \rho(\text{DMAGeI}_3) - \rho(\text{GeI}_3)$ where $\rho(\text{GeI}_3)$ are the electron density of GeI_3 framework with adding electron/f.u.: the residual electrons only locate at the DMA group. Therefore, our treatment of individual parts can restore the charge distribution in original DMAGeI_3 .

In short, our method keeps the accuracy of individual contributions to the maximum level. At least to our best knowledge, there isn't better method to partition the individual polarization since these two parts are not charge neutral. This point has been further clarified in the revised version.

Supplementary Fig. 18 | Electron density difference between DMAGeI_3 and extracted parts with holes/electrons added. a, Between the DMAGeI_3 and the pure DMA group. **b,** Between the DMAGeI_3 and the GeI_3 framework. The isosurfaces are set to $0.1 e/\text{\AA}^3$.

Comment 4: There is a certain relationship between the piezoelectric response and polarization of ferroelectrics. Given such a large P_s ($24.14 \mu\text{C}/\text{cm}^2$) of DMAGeI_3 , I suggest the authors to further measure its piezoelectric performance.

Response: We thank the reviewer for this suggestion. Following this suggestion, the piezoelectric test has been added to the revised supporting information (Supplementary Fig. 12). The macroscopic piezoelectric coefficient (d_{33}) was measured by a commercial piezometer (Piezotest, model: PM200) using the quasi-static method. At room temperature, the d_{33} of the DMAGeI_3 crystal was measured to be $3 \text{ pC}/\text{N}$, along the polar axis $[001]$ under a testing frequency of 110 Hz . Such poor piezoelectric response may be limited by its unipolar axis characteristics, which would be expected to be improved in multipolar axis ferroelectrics. The opportunities are vast for emerging hybrid germanium halide perovskites, and the ones

equipped with multipolar characteristics and outstanding piezoelectric performance will also be available in the near future.

The corresponding changes in the revised version:

Supplementary Fig. 12 | Photos of d_{33} meter. The maximum d_{33} of DMA GeI_3 is along the vicinity of the [001] direction of the crystal.

Comment 5: The SHG experiment is described in Supplementary Note 2. Please correct it in the manuscript.

Response: We thank the reviewer for reminding us of that, which has been fixed in the revised version.

Response to Reviewer #3

General Comment: The authors reported a quasi-one-dimensional perovskite DMAGeI_3 , which show excellent comprehensive ferroelectric performances comparable to commercial inorganic ferroelectric perovskites. Interestingly, the cooperation of ordering of polar cations and soft mode of $4s^2$ lone pairs in this perovskite results in large P_s of 24.14 uC/cm^2 , low E_c of 2.2 kV/cm , as well as strong SHG response. This new structural perspective would provide a feasible path to more superior ferroelectrics. Experimental characterizations and theoretical calculations are systematic and reliable. I think this work presents an exciting and significant advancement in ferroelectrics and this manuscript is suitable for publishing in Nature Communications before the following issues are addressed.

Response: We thank the reviewer for his/her careful reviewing and positive evaluation as “exciting and significant advancement”.

Comment 1: The reversibility of physical characteristics is closely related to thermal stability. How about the thermal stability (i.e., melting point) of the DMAGeI_3 crystal?

Response: Thanks for this suggestion. As the reviewer pointed out, the reversibility of physical characteristics is closely related to thermal stability. Following this suggestion, the thermal stability test has been added to the revised supporting information (Supplementary Fig. 2). The melting point of the DMAGeI_3 crystal is about 395 K . Moreover, the TGA curve has shown that the material begins to decompose above 400 K . In short, the DMAGeI_3 crystal shows excellent thermal stability at room temperature.

The corresponding changes in the revised version:

Supplementary Fig. 2 | TGA curve of DMAGeI_3 .

Comment 2: The SHG is performed in the ferroelectric phase; therefore, it is necessary to confirm the phase purity.

Response: Thanks for this precious comments. Following this suggestion, we have measured the phase purity of DMAGeI₃. In addition to original single-crystal XRD, we have measured the powder XRD (PXRD) of DMAGeI₃, which does not show any impurity peak and matches well with the simulated one based on the single-crystal data, indicating a high quality phase-pure sample (Supplementary Fig. 1). Furthermore, our PXRD pattern does not change after the sample exposes to air for 30 days, excluding possible secondary impurity phases.

The corresponding changes in the revised version:

Supplementary Fig. 1 | Powder X-ray diffraction patterns of DMAGeI₃ in different test conditions.

Comment 3: What about the SHG intensity vs. the particle size of crystals? The SHG responses of the tested material and KDP should be tested at the same particle sizes. Relevant phase matching tests are necessary.

Response: We thank the reviewer for this suggestion. Following this suggestion, the phase matching test has been added to the revised supporting information (Supplementary Fig. 7). The SHG intensities increase with the particle sizes and gradually reach saturated values. In the same particle size range of 300–450 μm, the DMAGeI₃ exhibits more than ten times the SHG strength of KDP (Fig. 2a). The corresponding particle sizes have been added in the caption of Fig. 2a in the revised manuscript.

The corresponding changes in the revised version:

Supplementary Fig. 7 | The particle size dependence of SHG intensity for both DMAGeI₃ and KDP under 1064 nm laser radiation.

Comment 4: The citation formats for some figures are inconsistent, such as "Supplementary Fig.8", and "Supplementary Fig.S12".

Response: Thanks for reminding us of that. We have revised these inappropriate citation formats and other errors.

Comment 5: The spontaneous polarization for DMAGeI₃ is resulting from the ordering of organic cations. According to the Curie-Weiss law, the type of ferroelectric phase transition needs to be further elucidated from the temperature-dependent dielectric curves.

Response: Thanks for this valuable suggestion. To further determine the type of ferroelectric phase transition, the curves of $1/\epsilon'$ vs temperature in the vicinity of T_c has been added to the revised supporting information (Supplementary Fig. 10). According to the Curie-Weiss law, the C_{para} and C_{ferro} are calculated to be 9664 and 17469 K at 1 kHz, respectively. The ratio of $C_{\text{ferro}}/C_{\text{para}}$ is 1.8 (smaller than 4), which discloses the characteristics of second-order ferroelectric phase transition.

The corresponding changes in the revised version:

Supplementary Fig. 10 | c, The fitting to Curie-Weiss law of dielectric anomalies in the vicinity of T_C at 1 kHz.

Comment 6: The authors mentioned the synthesis and structures of other halides in the supporting information. It is suggested to provide detailed experimental and conclusion information about this series of compounds.

Response: According to the reviewer's helpful suggestion, we have described the experimental synthesis in detail. The related contents have been added in the revision, which is beneficial for others to reproduce crystal growth.

The corresponding change can be found in Supplemental Note 1.

Comment 7: The authors are suggested to cite the related references on hybrid halide perovskites, such as "A Hybrid Antiperovskite with Strong Linear and Second-Order Nonlinear Optical Responses, *Angew. Chem. Int. Ed.*, 2022, e202211151" and "A Hybrid Halide Perovskite Birefringent Crystal, *Angew. Chem. Int. Ed.*, 2022, 61, e202202746".

Response: We thank the reviewer for bringing these two works to our attentions, which have been cited as references 45 and 21.

REVIEWERS' COMMENTS

Reviewer #1 (Remarks to the Author):

The authors have addressed all my queries and the paper can now be accepted as it is.

Reviewer #2 (Remarks to the Author):

Dear Editor

After careful review of the revised manuscript, I think our all concerns have been satisfactorily answered, so I suggest this manuscript can be accepted for publication now.

2023/4/23

Reviewer #3 (Remarks to the Author):

I am glad to see that my comments have been well addressed. I recommend publication of this manuscript.